# High-Temperature-Responsive Poplar lncRNAs Modulate Target Gene Expression via RNA Interference and Act as RNA Scaffolds to Enhance Heat Tolerance

**DOI:** 10.3390/ijms21186808

**Published:** 2020-09-16

**Authors:** Yuepeng Song, Panfei Chen, Peng Liu, Chenhao Bu, Deqiang Zhang

**Affiliations:** 1Beijing Advanced Innovation Center for Tree Breeding by Molecular Design, Beijing Forestry University, 35 Qinghua East Road, Beijing 100083, China; yuepengsong@bjfu.edu.cn (Y.S.); PanfeiChen@bjfu.edu.cn (P.C.); ackliup@163.com (P.L.); BuChenhao@bjfu.edu.cn (C.B.); 2National Engineering Laboratory for Tree Breeding, College of Biological Sciences and Technology, Beijing Forestry University, 35 Qinghua East Road, Beijing 100083, China; 3Key Laboratory of Genetics and Breeding in Forest Trees and Ornamental Plants, Ministry of Education, College of Biological Sciences and Technology, Beijing Forestry University, 35 Qinghua East Road, Beijing 100083, China

**Keywords:** long noncoding RNAs, high temperature, layered double hydroxide nanosheets, RNA structure, *Populus simonii*

## Abstract

High-temperature stress is a threat to plant development and survival. Long noncoding RNAs (lncRNAs) participate in plant stress responses, but their functions in the complex stress response network remain unknown. Poplar contributes to terrestrial ecological stability. In this study, we identified 204 high-temperature-responsive lncRNAs in an abiotic stress-tolerant poplar (*Populus simonii*) species using strand-specific RNA sequencing (ssRNA-seq). Mimicking overexpressed and repressed candidate lncRNAs in poplar was used to illuminate their regulation pattern on targets using nano sheet mediation. These lncRNAs were predicted to target 185 genes, of which 100 were *cis* genes and 119 were *trans* genes. Gene Ontology enrichment analysis showed that anatomical structure morphogenesis and response to stress and signaling were significantly enriched. Among heat-responsive LncRNAs, TCONS_00202587 binds to upstream sequences via its secondary structure and interferes with target gene transcription. TCONS_00260893 enhances calcium influx in response to high-temperature treatment by interfering with a specific variant/isoform of the target gene. Heterogeneous expression of these two lncRNA targets promoted photosynthetic protection and recovery, inhibited membrane peroxidation, and suppressed DNA damage in *Arabidopsis* under heat stress. These results showed that lncRNAs can regulate their target genes by acting as potential RNA scaffolds or through the RNA interference pathway.

## 1. Introduction

Due to global climate change, high-temperature stress has emerged as a major threat to plant growth and development [1]. Forest trees cover approximately 30% of the terrestrial land area worldwide and serve as foundation species that organize entire biotic communities and biogeochemical systems [2]. As perennial plants, trees possess a highly sophisticated genetic make-up so that they can respond to external stimuli [1]. A better understanding of the molecular responses of trees to high temperature will offer insights and options for tree improvement and ecosystem management.

Many studies have highlighted the importance of signaling involving various mitogen-activated protein kinases, phosphatases, and transcription factors as well as signaling molecules such as lipids, reactive oxygen species (ROS), and calcium (Ca^2+^) during high-temperature stress in annual plants [3]. Altered signal transduction leads to the differential expression of protein-coding genes, which results in changes in the intracellular milieu which play critical roles in adapting to stress [4]. More recently, long noncoding RNAs (lncRNAs) have been shown to play key roles in regulating gene expression under stress. They are derived from the transcription of divergent genomic regions and from splicing intermediates such as the circular intronic RNA produced by intron lariats, circular RNA produced by the back-splicing of exons, large intervening/intergenic noncoding RNA (lincRNA), natural antisense transcript (NAT), and promoter upstream transcript (PROMPT) [5]. According to their genomic location and targets, lncRNAs can be divided into cis-targeting (modulating genes proximal to the lncRNA locus) and trans-targeting (modulate genes that are far away from their origin) types. LncRNAs play critical roles in development, flowering, and biotic and abiotic stress responses by affecting the DNA methylation, messenger RNA (mRNA) transcription, splicing, export, stability, and translation of target genes [6]. In *Arabidopsis*, the lincRNA *COLD ASSISTED INTRONIC NONCODING RNA* interacts with polycomb repressive complex 2 to establish stable repressive chromatin at *FLOWERING LOCUS C* [7]. In addition to regulation through its primary sequence, lncRNAs also have specific secondary structures through which they can affect the DNA-binding activity of transcription factors by modifying transcription factor dimerization or trimerization [7,8]. In particular, R-loop chromatin is also enriched in the transcript location of lncRNAs in plants [9]. Overall, these studies suggest that, in plants, lncRNAs are present in vast numbers and regulate genes in *cis* and *trans*, consequently influencing diverse biological processes under normal and stress conditions. Thus, identification and functional analysis of lncRNAs that respond to high temperature is critically important for understanding not only gene regulation but also the molecular processes that are regulated by lncRNAs involved in plant development and stress responses.

Several previous studies identified lncRNAs in various plant species. Even after lncRNAs have been identified and their regulatory networks have been predicted, testing their functions remains challenging, particularly in large perennial plants. In this study, an attempt was made not only to identify high-temperature-responsive lncRNAs in poplar (*Populus simonii*) but also to functionally validate selected heat-responsive lncRNAs by transferring synthetic sense and antisense lncRNAs into live cells (mimicking overexpression and silencing). Recently, it was shown that delaminated lactate-containing layered double hydroxide nanosheets (LDS-NH) can be used to transport negatively charged biomolecules, such as DNA, RNA, and protein, into intact plant cells, thereby facilitating functional analysis [10,11]. This provides an efficient and convenient molecular tool to deliver synthetic single-stranded RNA mimicking lncRNA overexpression (sense lncRNA) or silencing (antisense lncRNA) in vivo.

In our study, 204 high-temperature-responsive lncRNAs were identified in *P. simonii*, which is an abiotic stress-tolerant poplar species. A total of 100 *cis* genes and 119 *trans* genes were predicted as potential targets for these lncRNAs. Based on annotation of the target genes, the potential functions were analyzed and the Gene Ontology (GO) term “response to stress” was significantly enriched. Functional analysis indicated that TCONS_00202587 can induce an abundance of *cis* target Potri.012G002800 through interaction with the upstream region of targets. By contrast, TCONS_00260893 can enhance Ca^2+^ influx in response to high-temperature treatment by interfering with a specific variant/isoform of the target gene. The overexpression of two lncRNA targets, Potri.012G002800 and Potri.017G089800, resulted in enhanced heat tolerance in *Arabidopsis*. These results shed light on the lncRNA transcript regulation pathway involved in heat tolerance of perennial plants.

## 2. Results

### 2.1. Characterization of High-Temperature-Responsive lncRNAs in P. simonii

To identify high-temperature-responsive lncRNAs in the leaves of *P. simonii*, stability of the photochemical efficiency of photosystem II (PSII) and antioxidant enzyme activity were used to optimize the high-temperature treatment. Temperature-dependent decline in the photochemical efficiency of PSII was used as a marker of the response of *P. simonii*’s to high temperature. We quantified the Fo under different temperatures. The results showed that the temperatures at which Fo reached its maximum (T_max_) and 50% of its maximum (T_50_) and at which the lines extrapolated from the slowly and quickly rising parts intersected in the temperature-Fo response curves (T_C_) were 47.1 °C, 45.2 °C, and 42.2 °C, respectively. Thus, those temperatures were used for the heat stress treatment. A time course analysis of transcripts encoding antioxidant enzymes such as peroxidase (POD), superoxide dismutase (SOD), catalase (CAT), and ascorbate peroxidase (APX) was performed. The highest elevation was observed after exposure to 42 °C for 6 h (Appendix A). Thus, we used this 6-h 42 °C heat stress treatment in the transcript analysis under heat stress.

Five 1-year-old clones of the same size (five biological repeats) were exposed to 42 °C for 6 h (high-temperature treatment). Five 1-year-old poplars grown in a plant growth chamber at ambient temperature (25 °C, 6 h treatment) were used for the control group. Sequencing of cDNA libraries generated from control and heat-treated leaf samples yielded approximately 156 and 149 million high-quality reads (mean values of three replicates), respectively. Of these, about 79% and 81% of reads from the control and heat-treated samples, respectively, could be mapped to the reference genome (Appendix A). These reads corresponded to 36,765 protein-coding genes and 13,612 putative lncRNAs in both libraries, as predicted using strict criteria (Figure 1A; Appendix A). Of the lncRNA loci identified, only 3007 were located in the intergenic regions; the remaining 10,605 overlapped with protein-coding genes. Among these overlapping lncRNA loci, the majority overlapped with exons, comprising 27.1% of all lncRNAs (Figure 1B).

Although there was wide variation (200–9970 bp) in the length of these lncRNAs, most ranged from 984 to 1129 bp (Figure 1C). Antisense lncRNAs ranged in length from 201 to 2862 bp (average = 464 bp), and most were 344–709 bp. Sense lncRNAs ranged between 201 and 4692 bp (average = 1073 bp), and most were 750–1008 bp (Figure 1C). The GC content of the antisense lncRNA loci ranged between 41.4% and 43.3%; this tended to be higher than that of the sense lncRNA loci (Figure 1D), suggesting that the GC contents of lncRNAs were strand-specific. The minimum free energy (MFE) of the high-temperature-responsive lncRNAs ranged between −347.7 kcal/mol and −46.10 kcal/mol, and the average MFE per bp was −0.19.

### 2.2. Identification of High-Temperature-Responsive lncRNAs and Their Potential Targets in P. simonii

A total of 3317 protein-coding genes were differentially regulated (upregulated or downregulated) in response to heat stress (Appendix A). Although 2118 lncRNAs were identified in heat-treated samples, the majority were only found in heat-treated samples with low transcript abundance. To increase confidence in our annotations, only lncRNAs found in both control and heat-treated samples were identified and then used to identify differentially regulated lncRNAs. This analysis resulted in the annotation of 204 lncRNAs as high-temperature-responsive lncRNAs (Appendix A). In general, lncRNAs were much lower in abundance than mRNAs (Appendix A), consistent with previous reports from diverse plant species such as *Arabidopsis*, *Setaria italica*, and *Medicago truncatula* [12,13,14].

RNAplex was used for identifying potential targets of 204 differentially expressed lncRNAs. Physical location (within ±2 kb) was used to predict potential *cis*-targets. Complementary sites were used to predict trans-target genes. This resulted in the prediction of 100 *cis* genes (genes within ± 2 kb of the lncRNA on the genome), 119 *trans* genes (based on complementary energy between lncRNA and distantly located protein-coding genes on the genome), and 42 genes regulated in both *cis* and *trans* (Appendix A). Among these genes, 58 *cis*- and 90 *trans*-regulated genes were found to be co-expressed with heat-responsive lncRNAs. GO analysis of the co-expressed genes revealed enrichment for eight GO terms, including “anatomical structure morphogenesis”, “response to stress” and “signaling” (Appendix A). Among the 185 genes, 3 and 2 were found to be involved in cell death and the cell cycle, respectively, indicating that heat-responsive lncRNAs might be involved in the DNA damage response (DDR) through regulation of the expression of these genes (Appendix A).

### 2.3. Effects of High-Temperature-Responsive lncRNAs on Target Gene Expression

The *cis* regulation of target genes of lncRNAs could be gauged simply by scrutinizing the RNA-seq profiles of genes neighboring (±2 kb) the lncRNAs. A strong negative correlation was found between the stress-responsive lncRNAs and their cis target genes (Figure 2A). For instance, under heat stress, the expression level of downregulated lncRNAs was significantly lower than that of neighboring *cis* target genes (Figure 2B), while the expression level of upregulated lncRNAs was significantly higher compared to cis target genes (Figure 2C). We also found similar negative correlations between heat-responsive lncRNAs and their *trans* target genes (Figure 2B,C). Interestingly, the abundance of lncRNAs overlapping protein-coding genes was significantly higher than that of the transcripts of their host genes (Figure 2D).

Regarding *cis* regulation, the distance between lncRNAs and their target genes, and their location (upstream or downstream of the lncRNAs) could be important factors influencing regulation. To evaluate this possibility, we compared the expression of *cis* genes at different distances from the lncRNAs. The *cis* genes showed distinct expression patterns according to the distance from the lncRNAs. For example, upstream of lncRNAs, regions that had a maximum ratio of lncRNA and neighboring gene transcript abundance were detected at −750 bp. Downstream of the lncRNA, regions that had a maximum ratio of lncRNA and neighboring gene transcript abundance were observed at 600 bp (Figure 2E).

### 2.4. LncRNAs TCONS_00202587 Transcribed from Proximal Promoter Sequences of a Downstream Gene Can Regulate Transcription by Acting as an RNA Scaffold

Dependent on predicted GO terms and pathways associated with the target genes of high-temperature-responsive lncRNA, we speculate it might play an important role in high-temperature signal transduction in poplar. To investigate the function of high-temperature-responsive lncRNAs, two predicted interactions were experimentally validated. The genomic location of lncRNAs and their target genes are important determinants of the transcriptional regulation of the latter. The functions of two high-temperature-responsive lncRNAs, TCONS_00202587 and TCONS_00260893, which are located on sense and antisense strands and target signal transduction-related genes, respectively, were analyzed in this study (Table 1). TCONS_00202587 is located 4.2 and 0.7 kb upstream from the transcriptional start site of Potri.012G002700 and Potri.012G002800, respectively (Appendix A). Potri.012G002800 was predicted as a *cis* target of TCONS_00202587 and had an opposite transcriptional direction to Potri.012G002700 (Appendix A). Potri.012G002800 is member of the *PROTEIN PHOSPHATASE 2C* gene family that functions in serine/threonine protein phosphatase. Potri.012G002800 is annotated as protein kinase activity. Both predicted targets are classified as signaling-related genes and had similar expression patterns to the lncRNA TCONS_00202587. To mimic the silencing and overexpression of lncRNAs, antisense RNA and sense RNA strands, respectively, were delivered into intact cell using LDH-lactate-NS. Silencing of lncRNA interference (lncRNAi) resulted in downregulation of both targets compared to the control group under high temperature. By contrast, overexpression of the lncRNA (lncRNAe) led to elevated transcript abundance of both targets under high temperature (Figure 3A–D). These results suggest that TCONS_00202587 might positively regulate its target genes. The predicted secondary structures of TCONS_00202587 included five stem-loop structures, which could serve as an RNA scaffold and could influence binding of transcription factors/complexes to the target locus.

The proposed RNA scaffold function of the secondary structure of lncRNA TCONS_00202587 was initially assessed in silico by deleting the five stem-loop structures individually. Based on the structure predictions, deleting either stem-loop1, stem-loop2, or stem-loop5 only removed the specific stem-loop structure without affecting the overall secondary structure. By contrast, without stem-loop3 or stem-loop4, the secondary structure of TCONS_00202587 was predicted to be significantly altered (Figure 4A,B and Appendix A), suggesting that stem-loop3 and stem-loop4 might play important roles in maintaining the specific RNA structure of TCONS_00202587.

To experimentally validate the importance of these five stem loops of TCONS_00202587 to the transcriptional regulation of target genes, the five stem loops were systematically deleted. Mutant lncRNAs lacking one of these stem-loops were synthesized and used in the assays. The expression of the two target genes was significantly affected in roots that received TCONS_00202587 via LDH, without stem-loop3 or stem-loop4 (Figure 4C). To confirm these results, the luciferase (LUC) reporter assay was performed. The promoters of Potri.012G002700 (1432 bp) and Potri.012G002800 (1398 bp) were cloned into pGreenII0800-LUC vectors to drive the *FIREFLY LUCIFERASE* (*LUC*) gene as a reporter. The *RENILLA LUCIFERASE* (*REN*) gene under control of the 35S promoter in the pGreenII 0800-LUC vector was used as the internal control. Mutant and full-length lncRNA of TCONS_00202587 were cloned into pBI121 vectors under control of the 35S promoter and were used as effectors. The results also showed that, among the five mutant lncRNAs, only co-expression of 35S::TCONS_00202587-∆stem-loop3 and TCONS_00202587-∆stem-loop4 with ProPotri.012G002800::Luc was associated with no change in luminescence intensity. It was confirmed that stem-loop 3 and stem-loop 4 were required for active transcription of Potri.012G002800 through interaction with its promoter (Figure 4D). However, the LUC assay indicated that none of the five mutant lncRNAs interacted with the promoter of Potri.012G002700 to induce luminescence, suggesting that TCONS_00202587 might not directly interact with the promoter of Potri.012G002700.

To identify the potential interactions between TCONS_00202587 and its target genes, an in silico approach was used to identify regions that could interact but had deletions that did not alter the overall secondary structure of the lncRNA. Using this criterion, two candidate regions (interaction region 1 (IR1) and IR2) were predicted (Figure 5). We generated mutants in which approximately 30% of the nucleotides were replaced in both regions to abolish interactions between the lncRNA and its targets (Figure 5A). An in vitro RNA-DNA binding assay revealed that the migration rates of putative binding sites interacting with TCONS_00202587 and TCONS_00202587 mutations in IR1 were not significantly altered. By contrast, putative binding sites interacting with TCONS_00202587 IR2 mutations showed significantly faster migration rates than those interacting with TCONS_00202587 (Figure 5B). Expression analysis showed that the transcription abundance of these two targets was not significantly altered by the overexpression of TCONS_00202587 with IR2 mutations compared to the control group (*p* < 0.01; Figure 5C). The LUC assay also showed that co-expression of 35S::TCONS_00202587 and 35S::TCONS_00202587 ∆IR1 with ProPotri.012G002800 led to an obvious increase in luminescence intensity (Figure 5D). In contrast, 35S::TCONS_00202587 ∆IR2 did not increase luminescence intensity. These data indicate that IR2 might be the region wherein lncRNA and its target genes interact and exert important effects on transcript regulation.

### 2.5. High-Temperature-Induced lncRNA TCONS_00260893 Affects the Expression of Specific Isoforms of Potri.017G089800 Transcripts

The lncRNA TCONS_00260893, the level of which is significantly upregulated under heat stress (Appendix A; Appendix A), was also investigated in terms of its interaction with its target gene. TCONS_00260893 has been predicted to target Potri.017G089800, which encodes cyclic nucleotide-gated ion channel 2 (CNGC2) that is also a known and important signal transduction gene. TCONS_00260893 and Potri.017G089800 exist as a partially overlapping natural antisense transcript (NAT). The interaction between these sense and antisense transcripts is predicted to induce RNA interference via the generation of siRNAs [15]. Therefore, we initially tested for the presence of NAT-derived siRNAs (nat-siRNAs) using Pln24NT [16]. This analysis identified three high-temperature-responsive siRNA clusters. Cluster HT01 has seven high-temperature-responsive siRNAs in the region of overlap between TCONS_00260893 and Potri.017G089800 (Appendix A; Appendix A), demonstrating that the expression of TCONS_00260893 and Potri.017G089800 generates siRNAs. To determine whether the interaction between TCONS_00260893 and Potri.017G089800 influences CNGC2 expression in vivo, lncRNA silencing and overexpression strategies were used.

LDHs are sheets formed from nanoparticles and can be used to deliver biomolecules, including DNA and RNA, into plant cells [10,17]. To knockdown endogenous lncRNA TCONS_00260893, we introduced a synthetic RNA complementary to TCONS_00260893 into plant cells using LDH-NS for lncRNAi (Appendix A). Roots allow easy observation of the translocation of LDH-NS. The expression patterns of TCONS_00260893 target genes are similar between roots and leaves (Appendix A). Poplar roots were selected for LDH-NS treatment in this study. Significant downregulation of transcripts of TCONS_00260893 targets was seen, although the transcript abundance of different isoforms of the target gene CNGC2 (Potri.017G089800.2, 3, 4, 6, and 8) significantly increased (Figure 3B). Our RNA-seq analysis detected six Potri.017G089800 isoforms, including 3′ alternatively spliced isoforms. Only isoform Potri.017G089800.7 was unaffected. This can be explained by the fact that this isoform lacks the target site for TCONS_00260893, unlike the other five isoforms.

### 2.6. TCONS_00260893 Enhances Ca^2+^ Influx under High-Temperature Treatment

Over the past two decades, cyclic nucleotide-gated ion channel (CNGCs) have been extensively studied in plants [18,19]. More recent studies have clearly established an important role for CNGCs in Ca^2+^ signaling under stress conditions [20]. The high-temperature-responsive lncRNA TCONS_00260893 identified in this study, which targets Potri.017G089800, is likely involved in Ca^2+^ flux and therefore in Ca^2+^ signaling during stress. To monitor Ca^2+^ flux in vivo in poplar exposed to high-temperature conditions, we used the noninvasive micro-test (NMT) method. To assess whether TCONS_00260893 is involved in stress-induced Ca^2+^ flux, transient overexpression or silencing of this lncRNA was induced and Ca^2+^ flux was compared to that in controls. As expected, the NMT assay showed a significant influx of extracellular Ca^2+^ in high-temperature-treated poplar (Figure 6A). Compared to the controls, poplar roots in which TCONS_00260893-lncRNAi was silenced displayed minimal changes in Ca^2+^ influx, overexpression of this lncRNA (TCONS_00260893-lncRNAe), and significantly enhanced Ca^2+^ influx in poplar roots under high-temperature stress (Figure 6A). The maximum influx of TCONS_00260893-lncRNAe into roots under heat stress was significantly higher than that into controls and TCONS_00260893-lncRNAi (Figure 6B). These results suggest that high-temperature-induced TCONS_00260893 might positively regulate Ca^2+^ influx under heat stress by controlling the relative proportions of *CNGC2* transcript variants in poplar.

### 2.7. Overexpressed lncRNA Targets Enhance Heat Tolerance of Arabidopsis

The role of two lncRNA targets heterologously overexpressed in *A. thaliana*, Potri.012G002800 and Potri.017G089800.7, in the heat stress response was investigated. Under heat stress (42 °C, 24 h), the survival rates of WT, Potri.012G002800-OE, and Potri.017G089800.7-OE were 33.3%, 88.8%, and 77.7%, respectively (Appendix A). The Fv/Fm ratio did not significantly differ among the WT, Potri.012G002800-OE, and Potri.017G089800.7-OE transgenic lines. However, the Fv/Fm ratio showed a smaller decrease in Potri.012G002800-OE and Potri.017G089800.7-OE compared to WT under heat stress (Figure 7A). Also, the Fv/Fm ratio of two transgenic *A. thaliana* lines recovered more rapidly after the removal of heat stress (42 °C, 2, 4 and 6 h) compared to the WT group. Quantum yield of PSII electron transport (φPSII) was significantly higher in the Potri.012G002800-OE and Potri.017G089800.7-OE transgenic lines. Compared to the WT group, the ΦPSII of two transgenic lines also slowly decreased under heat stress and rapidly increased after the removal thereof (Figure 7B), indicating that overexpressed Potri.012G002800 and Potri.017G089800.7 might protect the photosynthetic machinery and might promote recovery after stress.

Photosynthetic capacity analysis showed that 6 h heat stress led to the lowest Fv/Fm ratio and ΦPSII. Thus, this heat stress was used to examine the physiological changes in two transgenic lines (Figure 7C). Under heat stress, the activities of SOD, POD, and CAT in Potri.012G002800-OE and Potri.017G089800.7-OE transgenic lines were significantly higher than in the WT group. After removing heat stress, the activities of SOD, POD, and CAT in the Potri.012G002800-OE and Potri.017G089800.7-OE transgenic lines remained significantly higher than in the WT group. Regarding malondialdehyde (MDA) content, MDA was significantly induced under heat stress in the WT group and remained at a higher level compared to that in Potri.012G002800-OE and Potri.017G089800.7-OE transgenic lines after removal of the heat stress. These data indicated that the higher antioxidant enzyme activity in the Potri.012G002800-OE and Potri.017G089800.7-OE transgenic lines prevented inhibition of membrane peroxidation.

To determine whether overexpressed of Potri.012G002800 and Potri.017G089800.7 had the DDR under heat stress, the expression profiles of six heat-responsive DDR-related genes were examined (Figure 7D) [21]. Among six candidate genes, the *MEIOTIC RECOMBINATION11* (*MRE11*) and *ATR-INTERACTING PROTEIN* (*ATRIP*) genes were sensors of the DDR. The transcript abundance of the *MRN* and *ATRIP* genes in transgenic *Arabidopsis* was significantly higher than that in the control group, suggesting that these DDR signal-sensing genes were induced by Potri.012G002800 and Potri.017G089800.7 overexpression. *ATAXIA TELANGIECTASIA MUTATED* (*ATM*) and *RAD3-RELATED* (*ATR*) and *SUPPRESSOR OF GAMMA RESPONSE1* (*SOG1*) were transducers in the DDR. The gene transcript abundance of the transducers was significantly higher in two transgenic samples than in the control group, with the exception of the ATR gene in Potri.012G002800-overexpressed transgenic *Arabidopsis* (which showed the same expression level in the control group after removal of heat stress). *CYCB1* and Replication factor-A protein 1 (*RPA1E*), as effectors of the DDR, were induced in two transgenic *Arabidopsis* samples, although RPA1E did not show differences in transcript abundance between the control and transgenic *Arabidopsis* samples under normal conditions. These results indicated that the activated transducers and effectors differed between the two transgenic *Arabidopsis* samples. In summary, different DDR pathways were activated by Potri.012G002800 and Potri.017G089800.7 overexpression, which might play an important role in suppressing DNA damage and in enhancing heat tolerance in plants.

## 3. Discussion

### 3.1. LncRNAs Function as Regulators of Temperature-Responsive Transcription

Our analyses revealed that the overall abundance of lncRNAs was almost six-fold lower compared to the transcripts of protein-coding genes; these findings are consistent with a previous report [22]. However, heat-responsive lncRNAs were expressed at high levels relative to the overall lncRNA population; their levels were only about two-fold lower than those of protein-coding transcripts, suggesting that temperature-responsive lncRNAs are more sensitive to heat stress than protein-coding genes. The majority of plant lncRNAs are expressed at low levels according to tissue analyses; however, their levels could be higher in certain cell types [12]. The predicted GO terms and pathways associated with the target genes of high-temperature-responsive lncRNA include response to stress and signaling, indicating that lncRNAs may play an important role in high-temperature signaling in poplar. LncRNA target gene annotation and candidate gene overexpression analysis indicated that sensors, transducers, and effectors of the DDR pathway were involved in lncRNA-mediated transcription networks, in turn implying that lncRNAs play a crucial role in DDR regulation under heat stress.

MFE is always negatively associated with the stability of RNA secondary structures [23]. Our results showed that the MFE and MFE per bp of high-temperature-responsive lncRNAs were substantially lower than those of osmotic stress-responsive lncRNAs in poplar [24]. In bacteria, mRNA stem-loop structures can be reshaped by high temperature, facilitating ribosome binding and translation [25]. It appears that more stable lncRNAs are required for transcript regulation under heat stress. Sense and antisense RNA pairs might induce RNA interference. Our computational and expression analyses revealed that TCONS_00260893 is transcribed from antisense strands and plays a negative regulatory role with respect to the expression of its target genes. Coupled with siRNA analysis, this suggested that these antisense RNAs can regulate target gene expression through endogenous RNA interference.

### 3.2. Specific Secondary Structures of lncRNAs Are Essential for Transcript Regulation

RNA structures often play important roles in processes ranging from ligand sensing to the regulation of translation, polyadenylation, and splicing [26]. In *Arabidopsis*, many conserved structural motifs of lncRNAs have been found to be responsible for stress-response functions (e.g., an adenylate-uridylate (AU)-rich stem-loop responding to cold) [27]. Under osmotic stress, PROMPT_1281 can act as a carrier of MYB transcription factors through its secondary structure and thus can regulate the expression of target genes [24]. It was shown that, without the region interacting with targets, PROMPT_1281 establishes a concentration gradient that increases the probability of its interaction with targets near its transcription site that share common motifs. By contrast, this study found that TCONS_00202587 is a positive regulator of its targets. In vitro experiments showed that TCONS_00202587 can hybridize with the promoter sequence of *trans* targets. Expression analysis indicated that the secondary structure of TCONS_00202587 without stem-loop3 and stem-loop4 was significantly changed and unable to induce target expression. This finding indicated that TCONS_00202587 binding to upstream sequences of its targets utilizes its secondary structure to regulate target transcription. Our results revealed the IR of lncRNAs and the DNA sequence are also essential for transcriptional regulation, which led us to hypothesize that features and functions of the IR affect lncRNA secondary structure.

Conserved motifs (primary sequences or secondary structures) are essential for RNAs to be bound and regulated by RNA-binding proteins [28]. This suggests that specific spatial arrangements are essential for lncRNAs to be bound by RNA-binding proteins. Our results support the idea that specific secondary structures of lncRNAs TCONS_00202587 are important for enhancing adjacent gene expression. LncRNA can regulate target gene expression through modifying transcription factor dimerization, promoting transcription factor phosphorylation [7], or controlling transcription factor nuclear localization [8]. However, the element that interacts with TCONS_00202587 still needs to be investigated, perhaps through RNA pulldown analysis. RNA structures are sensitive to environmental changes [29]. In vitro and in vivo analysis of RNA secondary structures indicated that RNAs might undergo conformational changes under stress [30]. Especially under heat stress, asymmetric changes within the cell and across organs [31] might trigger lncRNAs to remodel their secondary structures. Although in vivo click-selective 2′-hydroxyl acylation and profiling experiments can be used to analyze the genome-wide RNA secondary structure in vivo or in vitro, understanding the stress-responsive tissue-specific expression patterns of lncRNAs and the lower abundance of their transcription factors is still a challenge in secondary structure profiling.

### 3.3. Temperature-Responsive lncRNAs TCONS_00260893 Mediate Ca^2+^ Signaling Transduction in Poplar

In plants, cytosolic Ca^2+^ elevation facilitates the perception of external and internal cues, which in turn results in altered gene expression and cellular function [32]. Although Ca^2+^ influx is the earliest response in the induced signaling cascades, the mediating channels are mostly unknown, except the CNGC family of proteins and glutamate receptor homologs [33,34]. Our results showed that Ca^2+^ rapidly influxes into the cytoplasm in response to high temperature. This indicated that Ca^2+^, as an important secondary messenger, is also important for the perception of high temperatures and signaling transduction. *CNGC2* encoding CYCLIC NUCLEOTIDE GATED CALCIUM CHANNEL 2, which is a component of cyclic nucleotide gated Ca2+ channels, has been identified as the primary thermosensor of land plant cells [35]. In this study, Potri.017G089800 was predicted as a *cis* target for the lncRNA TCONS_00260893. All isoforms of Potri.017G089800 except Potri.017G089800.7 were suppressed when TCONS_00260893 was upregulated under high temperature. Alternative splicing of mRNA produces a wide variety of differently spliced RNA transcripts, which significantly increases the complexity of the transcriptome. Reliable large-scale mass spectrometry-based proteomics analyses indicated that most human genes have a single main protein isoform [36]. Our results showed that Potri.017G089800.7 contained core functional domains of CNGC2 as its main RNA isoform, suggesting that Potri.017G089800.7 transcription alone maintains Ca^2+^ influx under heat stress. These observations strongly suggest that TCONS_00260893 can regulate the relative proportions of CNGC2 transcript variants in poplar to promote Ca^2+^ influx under heat stress. Our results revealed that Potri.017G089800.7 is the shortest transcript among the eight CNGC2 isoforms; it is four exons shorter than the full-length (1850 bp) CNGC2. It was proposed that almost 70% of alternative isoforms had lost more than 60 residues, which could affect protein domains and, in turn, interactions with their protein partners [37]. Further studies should determine whether this is applicable to CNGC2 isoforms.

## 4. Materials and Methods

### 4.1. Plant Materials and Treatment

Ten annual *P. simonii* clones that were grown in pots under natural light conditions (1250 µmol m^−2^s^−1^ of photosynthetically active radiation) and photoperiod at 25 ± 1 °C and 50 ± 1% relative humidity, with a 16 h day/8 h night regime in an air-conditioned greenhouse were used in our study. The temperature-dependent decline in PSII was used as a marker of the response of *P. simonii*’s to high temperature according to Knight and Ackerly (2002) [38]. We quantified the minimal fluorescence (Fo) under different temperatures using MultispeQ (Photosynq, East Lansing, MI, USA). Five 1-year-old clones of the same size (five biological repeats) were exposed to 42 °C for 6 h (high-temperature treatment). Five 1-year-old poplar grown in a plant growth chamber (KRQ-400P; Keelrein Inc., Shanghai, China) at ambient temperature (25 °C, 6 h treatment) were used for the control group. To avoid drought stress during heat stress treatment, the plants were irrigated with water before treatment. The control and heat stress treatment groups were maintained under the same irradiance and relative humidity (1250 μmol·m^−2^·s^−1^ of photosynthetic photon flux, 50 ± 1% relative humidity). At the end of the stress treatment, sixth mature leaves were collected from untreated and heat stress-treated plants, immediately frozen in liquid nitrogen, and stored at −80 °C for RNA analysis.

### 4.2. RNA Sequencing and Prediction of lncRNAs

Total RNA was isolated from the leaves via a modified cetyl-trimethylammonium bromide (CTAB) method and used to construct a strand-specific cDNA or small RNA library. The library construction procedure is detailed in the Appendix A [39]. The total RNAs were sequenced using the HiSeq 2000 sequencing platform (Illumina, San Diego, CA, USA). The total number of reads and their mapping results are shown in Appendix A. The RNA sequencing (RNA-seq) data (three biological repeats) reported here are available from the National Center for Biotechnology Information (accession No. PRJNA357670).

Low-quality reads were discarded, and the adapter sequences from the remaining reads were trimmed to obtain clean reads. These reads were mapped to the *P. trichocarpa* reference genome (version 3.0), which served as the reference for *P. simonii*, using HISAT2 (version 2.0.5) [40]; mapping of up to three mismatches was allowed. To identify lncRNA candidates, three filters were used in this study. First, the length of transcriptional units (TUs) had to be longer than 200 base pairs (bp). Second, the longest open reading frame (ORF) of the TU had to be smaller than 300 bp (the longest ORF predicted by OrfPredictor; http://proteomics.ysu.edu/tools/OrfPredictor.html) [41]. Both sense and antisense strands of the TUs were used for prediction. Third, to be more confident in our predictions, only lncRNAs that appeared in both treatments (control and heat stress) were considered. The coding potential calculator (CPC) and coding-noncoding index (CNCI) (CPC score < 0 and CNCI < 0) were used to assess the protein-coding potential of a transcript [42,43]. Pfam Scan (version 1.3) was used to identify the occurrence of any known protein family domains documented in the Pfam database (release 32) [44].

### 4.3. Predication of Small Interfering RNA

Small RNA data were used only for predicting and identifying small interfering RNAs (siRNAs). FASTX (fastx_toolkit-0.0.13.2) was used to filter low-quality reads and adapters. The siRNAs were predicated by Pln24NT (v1.0). The 24-nucleotide siRNA read sequences were retrieved and aligned to microRNAs, and noncoding RNAs from the Rfam database were removed with Perl scripts [16]. Non-redundant 24-nucleotide siRNA sequences were mapped to the *P. trichocarpa* reference genome (version 3.0) using Bowtie [45]. A maximum of one mismatch (-v 1) was allowed, and the best alignments of reads with no more than 50 hits (-a -m 50 --best --strata) were reported. SiRNA cluster calling was carried out using ShortStack (version 3.3) with a minimum coverage of 10 siRNA reads 24 nucleotides in length. Genomic regions associated with siRNA clusters are referred to as siRNA loci. Identified 24-nucleotide siRNA clusters were merged if present in a window of 150 bp [46] to generate the final set of 24-nucleotide siRNA loci.

### 4.4. Structural Motifs and Target Prediction for Heat-Responsive lncRNAs

Conserved sequence motifs in a group of lncRNAs were searched for using MEME Suite (http://meme-suite.org/tools/meme) [47]. Depending on the common RNA binding protein binding domain, the motif width was constrained to 4–12 nucleotides. The significance threshold was set to an E-value of 0.05. Then, the predicted conserved sequence motif was used for GO term enrichment analysis performed with the GOMo program of the MEME suite (ver. 4.12.0; http://meme-suite.org/tools/gomo). The significance threshold was set to a q-value of 0.01. The *A. thaliana* genome was used as the reference genome.

For the lncRNAs identified by the GO term, we predicted the significantly conserved structural motifs (*p* < 0.001) using RNApromo (https://genie.weizmann.ac.il/pubs/rnamotifs08/rnamotifs08_predict.html) [48]. Then, the predicted motifs in all temperature-responsive lncRNAs were searched for similar structural motifs. The RNApromo program “rnamotifs08_motif_match.pl” was used for these searches, with a likelihood score > 0 and a false-positive rate of 0.05. By using genome annotation and a genome browser [49], potential *cis* target genes that were physically close to the lncRNAs (within ± 2 kb) were predicted. Regarding potential *trans* genes in poplar, BLAST was used to identify sequences that were complementary to the lncRNA, with the e-value set to < 1 × 10^−5^ and identity to ≥95%. Then, RNAplex was used to calculate the complementary energy between two sequences to facilitate identification of potential trans-acting genes (e = −60) [50].

### 4.5. Quantitative PCR Analysis

To validate the identified differentially regulated lncRNAs, quantitative PCR (qPCR) analysis of 40 heat-responsive lncRNAs was performed. The primer sets used for validation are listed in Appendix A. The qPCR analysis was performed using the ABI StepOne Plus instrument (Applied Biosystems, Foster City, CA, USA). Sample cycle threshold (Ct) values were determined and standardized relative to the endogenous control gene (ACTIN), and the 2–ΔΔCT method was used to calculate relative changes in gene expression based on the qPCR data [51]. The RNA-seq data were also validated by qPCR analysis of candidate mRNAs and lncRNAs with wide variation in expression patterns. A melting curve was used to check the specificity of each amplified fragment. For all reactions, triplicate technical and biological repetitions were performed for each individual. After amplification, the PCR products were sequenced to check the specificity of the primer sets. We found a significant correlation between the qPCR and RNA-seq transcript abundance data (*r* = 0.72, *p* < 0.001), indicating the reliability of the RNA-seq data (Appendix A).

### 4.6. Differential Expression Analysis

Cuffdiff (version 2.1.1; Department of Stem Cell and Regenerative Biology, Harvard University, Cambridge, MA, USA) was used to calculate fragments per kilobase of exon per million fragments mapped (FPKM) values [52]. The FPKM values of genes and lncRNAs were computed by summing the FPKM values of the transcripts in each transcript group. Cuffdiff provides statistical routines for determining differential expression in transcript data using a model based on the negative binomial distribution [52]. Differential expression analysis of two conditions or groups was performed using the DESeq R package (ver. 1.8.3; European Molecular Biology Laboratory Heidelberg, Germany) [53]. The *p*-values were adjusted using the method of Benjamini and Hochberg (1995) [54]. Differences in mRNA levels were considered statistically significant at a fold change > 2 or < 0.5 (*p* < 0.01, *q* < 0.05). Pearson’s correlation and complete linkage clustering were used for hierarchical clustering based on gene expression patterns [55]. All genes were annotated using the PopGenIE database [56]. The *A. thaliana* genome (TAIR10.1) was used as the reference genome.

### 4.7. Treatment with LDH–lncRNA Conjugates

Bulk Mg-Al-lactate LDH was synthesized using a coprecipitation method and delaminated in water to form nanosheets. The delaminated LDH-lactate is denoted as LDH-lactate-NS, with a final concentration of 1 mg/mL. To mimic the silencing and overexpression of lncRNAs, antisense RNA and sense RNA strands, respectively, were synthesized with fluorescein 5-isothiocyanate (FITC) at the 5′ end (Appendix A) (Sangon Biotech, Shanghai, China). These lncRNAs were dissolved in distilled water to a concentration of 1 mg/mL [57]. The LDH-lactate-NS colloid in Murashige and Skoog (MS) medium was added dropwise to lncRNAs at a volume ratio of 3:1, followed by gentle mixing. RNase inhibitor was added to a final concentration of 0.4 U/μL. The mixture was incubated for 1 h to form the LDH-lactate-NS–lncRNA conjugate. Then, 100 μL LDH-lactate-NS–lncRNA conjugate was added to liquid MS medium so that the RNA could be transported into the plant roots (Appendix A). Poplar roots were dipped in liquid MS medium with 0.2 μg/μL LDH-lactate-NS–lncRNA conjugate. After incubation at room temperature for 3 h, the roots were rinsed several times with standard growth medium, observed under a fluorescence microscope (BX61; Olympus, Tokyo, Japan), and stored at −80 °C for expression analysis. For fluorescence analysis, FITC was excited with a 488-nm laser, and the emission fluorescence was collected at wavelengths between 500 and 550 nm.

### 4.8. Dual Luciferase Assay

As a reporter, the promoters of Potri.012G002700 (1432 bp) and Potri.012G002800 (1398 bp) were cloned into pGreenII0800-LUC vectors to drive the firefly luciferase (LUC) gene. The *REN* gene under control of the 35S promoter in the pGreenII 0800-LUC vector was used as the internal control. Mutant and full-length lncRNA of TCONS_00202587 were cloned into pBI121 vectors under control of the 35S promoter and were used as effectors. The effector and reporter vectors were co-transformed into leaves of *P. simonii* (sixth mature leaves, 72 h), and dual luciferase activity was detected using a modified version of the method of He et al. (2013) [58]. Dual luciferase assay was performed using the Glomax20/20 luminometer (Promega, Madison, WI, USA) according to the manufacturer’s instructions, and LUC/REN ratios were calculated. The primers used for these analyses are listed in Appendix A.

### 4.9. Extracellular Ca^2+^ Flux Monitoring

Roots (1 cm) of tissue-cultured poplar seedlings, including control plants, were treated with LDH-lactate-NS–lncRNA conjugates and monitored for Ca^2+^ flux with a NMT technique (NMT-YG-100; Younger USA LLC, Amherst, MA, USA) as described by Sun et al. (2010) [59]. Pre-pulled and silanized glass micropipettes 4–5 μm in diameter (Xuyue Science and Technology Co. Ltd., Beijing, China) were back-filled with backfilling solution (Ca^2+^ microelectrodes: 100 mM CaCl_2_) to within 1.0 cm of the tip. Then, the micropipettes were front-filled with 15 μm columns of selective liquid ion exchange cocktails (Fluka 21048; Fluka Chemie GmbH, Buchs, Switzerland) [60]. To create an electrical contact with the electrolyte solution, an Ag/AgCl wire electrode holder was inserted into the back of the electrode. DRIREF-2 (World Precision Instruments, Sarasota, FL, USA) was chosen as the internal reference electrode. Ion-selective microelectrodes for the target ions were calibrated by Ca^2+^ standard solution (0.1, 0.5, and 1.0 mM) before measurement. The Ca^2+^ concentration in the measuring solution was 0.2 mM. After a period of measurement at room temperature, high-temperature MS solution (40 °C) was added quickly to the container around the roots and Ca^2+^ flux was immediately measured. These analyses were performed independently with three replicates. The flux rate was calculated on the basis of Fick’s law of diffusion:J = −D (dc/dx)

In this formula, J represents the ion flux in the x direction, D represents the ion diffusion coefficient in a particular medium, dc is the ion concentration difference, and dx represents the microelectrode movement between two positions.

### 4.10. Overexpressed Heat-Responsive lncRNA Targets in A. thaliana

After disinfection and vernalization, *A. thaliana* (Col-0) seeds were sown on 1/2 MS medium in a growth chamber (22 °C/16 h light and 18 °C/8 h dark cycle). After 4 weeks, seedlings were transplanted into soil in a growth chamber (22 °C/8 h light and 18 °C/16 h dark cycle) for 1 week and the growth conditions were changed to a 22 °C/16 h light and 18 °C/8 h dark cycle to obtain adult plants.

To construct the lncRNA target overexpression vector, the coding sequences of Potri.012G002800 and Potri.017G089800.7 were amplified from the *P. simonii* clone ′QL9′cDNA library, cloned into the pCXSN vector driven by the 35S promoter, and introduced into Agrobacterium GV3101. Agrobacterium inoculated with the overexpression vector was used for infection of wildtype (WT) *A. thaliana* (Col-0) plants using the floral dip method. Heterologous overexpressing transgenic plants were obtained using the Ag. tumefaciens-mediated floral dip transformation method and hygromycin-containing (50 mmol/L) medium and progressed to the T2 generation. Homozygous transgenic lines were used for subsequent experiments. To grow transgenic *A. thaliana* seedlings, seeds were surface-sterilized and planted on 0.8% agar-solidified MS medium. After incubation for 4 days at 4 °C in darkness, the seeds were exposed to continuous white light at an intensity of 120 mmol/m^2^/s and incubated in a 22 °C/16 h light and 18 °C/8 h dark cycle for 2 weeks.

### 4.11. Measurement of Chlorophyll Fluorescence, Physiological, and Biochemical Characteristics of Transgenic Arabidopsis

Two-week-old (14 days) WT and transgenic *Arabidopsis* samples were exposed to constant high-temperature conditions (42 °C, 70% relative humidity) for 2, 4, and 6 h. Samples grown at constant room temperature (25 °C, 70% relative humidity) were used as the control group. Each treatment group, including the control group, contained three biological replicates. To assess the recovery of chlorophyll fluorescence under heat stress, each treatment group was returned to room temperature conditions after 1 h, and the variable fluorescence (Fv)/maximum fluorescence (Fm) ratio and φPSII were measured using MultispeQ (Photosynq). To determine the survival rate of WT and two transgenic *Arabidopsis*, all samples were exposed to 42 °C for 24 h. Relative humidity was set to 60% ± 1 and held constant during the measurements. Chlorophyll fluorescence images were obtained using the Chlorophyll Fluorescence System (Heinz-Walz Instruments, Effeltrich, Germany) according to the manufacturer’s instructions. Malondialdehyde (MDA), peroxidase (POD), superoxide dismutase (SOD), catalase (CAT), and ascorbate peroxidase (APX) activities were measured by absorption photometry using a spectrophotometer, according to the method described by Song et al. [61,62].

### 4.12. Statistical Analysis

Significant differences in physiological characteristics, biochemical characteristics, and expression data among the different groups according to heat stress duration were determined using Fisher’s least significant difference (LSD) test. Differences were considered statistically significant when *p* < 0.01.

## 5. Conclusions

In plants, lncRNAs appear to participate in diverse biological processes including development and stress responses. In this study, high-temperature-responsive lncRNAs, TCONS_00202587, and TCONS_00260893, were shown to regulate their targets via RNA interference or to act as RNA scaffolds, thereby promoting photosynthetic protection and recovery, inhibiting membrane peroxidation, and suppressing DNA damage in plants under heat stress. This study provides new insight into transcript regulation of lncRNA in poplar heat tolerance and lays the foundation for further RNA structures function research in perennial plants.

## Figures and Tables

**Figure 1 ijms-21-06808-f001:**
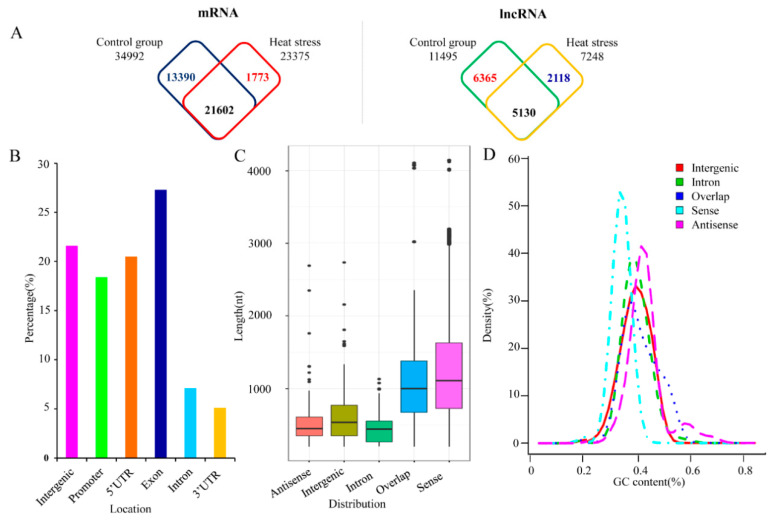
Characteristics of high-temperature-responsive noncoding RNAs in poplar: (**A**) number of mRNAs (left) and long noncoding RNAs (lncRNAs) (right) expressed in response to high temperature, (**B**) percentage of expressed lncRNAs that produce transcripts from different locations in the poplar genome, (**C**) length of expressed lncRNAs with transcripts from different locations in the poplar genome, and (**D**) GC content of expressed ncRNAs that produce transcripts from different locations in the poplar genome. Antisense and sense represent lncRNA transcripts from antisense and sense strands of DNA double strands, respectively. Intergenic represents lncRNA transcripts from intergenic regions of the genome. Intron represents transcripts from intron regions of the gene. Overlap represents lncRNA transcripts in the region overlapping with 5′ UTR or 3′ UTR.

**Figure 2 ijms-21-06808-f002:**
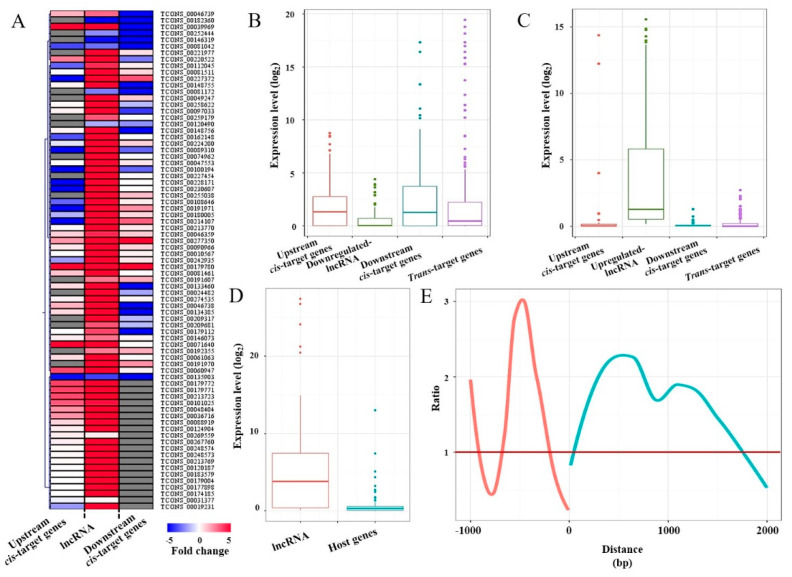
Expression of high-temperature-responsive lncRNAs and neighboring genes: (**A**) heatmap of high-temperature-responsive lncRNAs and neighboring genes (fold change > 2, *p* < 0.01). Grey bands represent cases where no gene was detected. (**B**) Expression of downregulated lncRNAs and neighboring genes: horizontal lines represent median values, vertical lines represent minimum and maximum values, and dots represent outliers. (**C**) Expression of upregulated lncRNAs and neighboring genes: horizontal lines represent median values, vertical lines represent minimum and maximum values, and dots represent outliers. (**D**) Expression of lncRNAs and host genes: horizontal lines represent median values, vertical lines represent minimum and maximum values, and dots represent outliers. (**E**) Trend in lncRNA and neighboring gene transcript abundance over distance.

**Figure 3 ijms-21-06808-f003:**
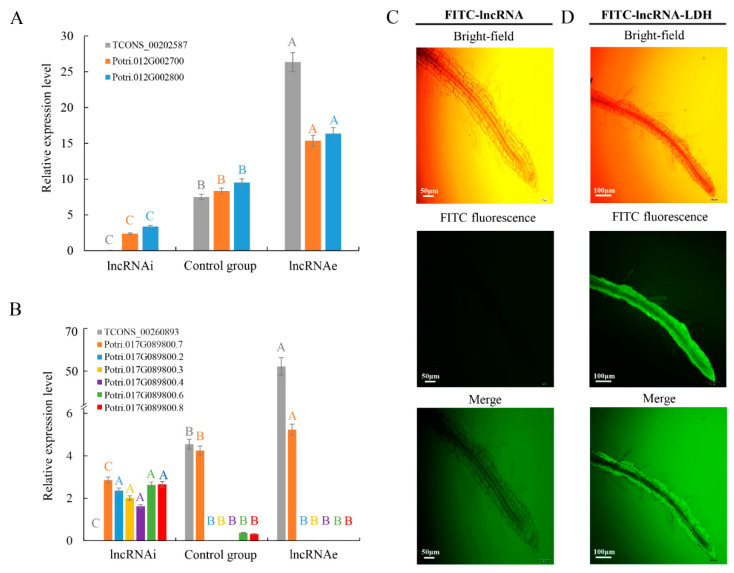
Expression of lncRNA and targets under lncRNAi and lncRNAe treatment: (**A**) expression pattern of TCONS_00202587 and its *cis* target genes (Potri.012G002700 and Potri.012G002800) and (**B**) expression of TCONS_00260893 and transcripts of different targets. Relative transcript levels were calculated by real time PCR with ACTIN as the standard. Data are mean ± SE of three separate measurements. Error bars represent standard error. Significantly different genes among the control group, lncRNAi, and lncRNAe were determined using Fisher’s least significant difference (LSD) test. Different colored letters on error bars indicate significant differences in the genes corresponding to that color among the groups at *p* < 0.01. (**C**) Fluorescence microscopy of intact poplar root cells after 3 h treatment with fluorescein 5-isothiocyanate (FITC)-lncRNAs and washing. Scale bars = 50 μm. (**D**) Fluorescence microscopy of intact poplar root cells after 3 h treatment with FITC-lncRNAs-LDH and washing. Green represents the fluorescence of LDH-lactate-NS-lncRNA-FITC from the cytosol of poplar root cells, indicating that LDH-lactate-NS delivered synthetic lncRNAs into intact roots. Scale bar = 100 μm.

**Figure 4 ijms-21-06808-f004:**
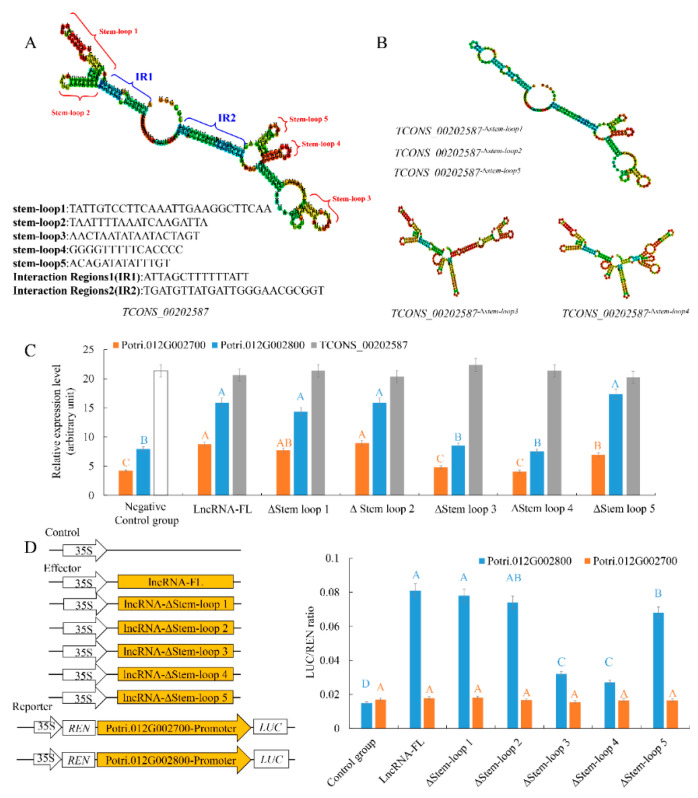
The secondary structure of TCONS_00202587 regulates the transcript levels of its targets. (**A**) Schematic diagram of the secondary structure and sequences of TCONS_00202587: Stem-loops 1–5 represent the typical secondary structure of TCONS_00202587. IR 1 and IR2 represent two potential interaction regions binding with the promoter of targets. (**B**) Schematic diagram of secondary structure mutants: TCONS_00202587-∆stem-loop3 and TCONS_00202587-∆stem-loop4 show significant changes in their secondary structures. (**C**) Target expression patterns under LDH-NP binding with TCONS_00202587 and its secondary structure mutant: lncRNA-FL represents full-length lncRNA. ∆stem-loop 1–5 represents deletion of stem-loops 1–5, respectively. Negative control group represents LDH-NPs binding with a random lncRNA sequence. (**D**) Transcriptional activation of TCONS_00202587 and its secondary structure mutant on two targets: The data are mean ±standard deviation (SD) of separate transfections (*n* = 3). Significantly different genes among the control group, LncRNA-FL, ∆stem-loop 1, ∆stem-loop 2, ∆stem-loop 3, ∆stem-loop 4, and ∆stem-loop 5 were determined using the LSD test. Different colored letters on error bars indicate significant differences in the genes corresponding to that color among groups at *p* < 0.01.

**Figure 5 ijms-21-06808-f005:**
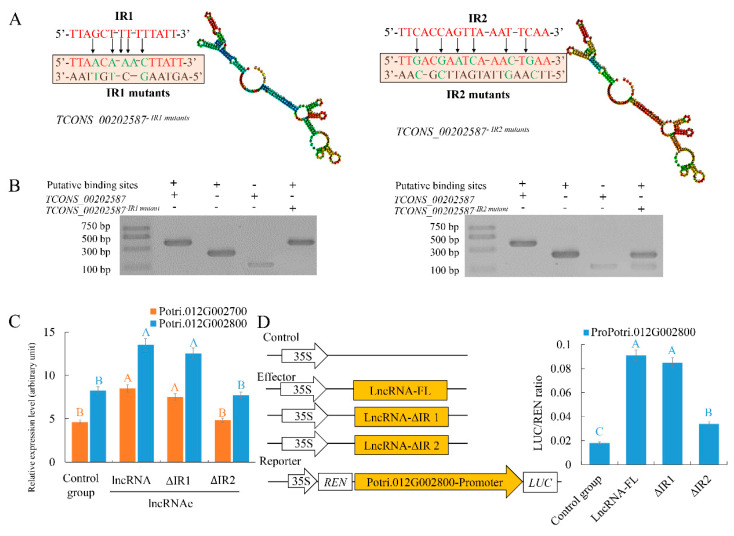
Verification of the interaction of TCONS_00202587 with its target genes Potri.012G002700 and Potri.012G002800: (**A**) mutant sequence of putative interaction regions and (**B**) agarose gel electrophoresis of lncRNA and putative binding sites. Lane 1 was loaded with RNA containing the putative binding sites and synthetic TCONS_00202587 RNA. Lane 2 was loaded with putative binding sites. Lane 3 was loaded with synthetic TCONS_00202587 RNA. Lane 4 was loaded with putative binding sites and synthetic TCONS_00202587-I12-mutant/TCONS_00202587-IR2-mutant sequences. (**C**) Target expression patterns under LDH-NP binding with TCONS_00202587 and its putative interaction region mutant: control group represents non LDH-NPs binding with lncRNA. lncRNA-FL represents full-length lncRNA. ∆IR 1 and 2 represent deletion of putative interaction regions 1 and 2, respectively. (**D**) Transcriptional activation of TCONS_00202587 and its interaction region mutant on two targets: The data are mean ± SD of separate transfections (*n* = 3). Significantly different genes among the control group, LncRNA, ∆IR1, and ∆IR2 were determined using Fisher’s LSD test. Different colored letters on error bars indicate significant differences in the genes corresponding to that color among the groups at *p* < 0.01.

**Figure 6 ijms-21-06808-f006:**
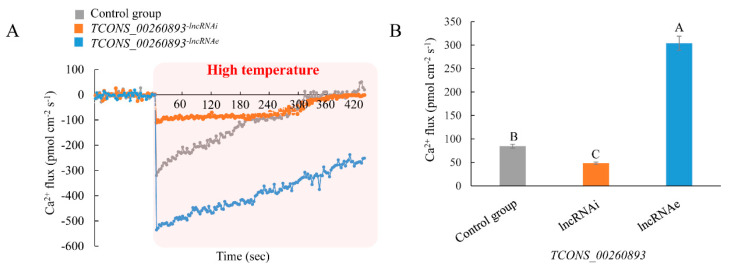
Ca^2+^ influx under high-temperature treatment in poplar: (**A**) NMT measurements show extracellular Ca^2+^ influx under high-temperature treatment in live roots treated with different LDH–lncRNA conjugates (*n* > 5). (**B**) Maximum Ca^2+^ influx of different samples: values are expressed as mean ± SD, *n* > 5. All RNAs were delivered into intact roots using LDH. Different letters on error bars indicate significant differences at *p* < 0.01.

**Figure 7 ijms-21-06808-f007:**
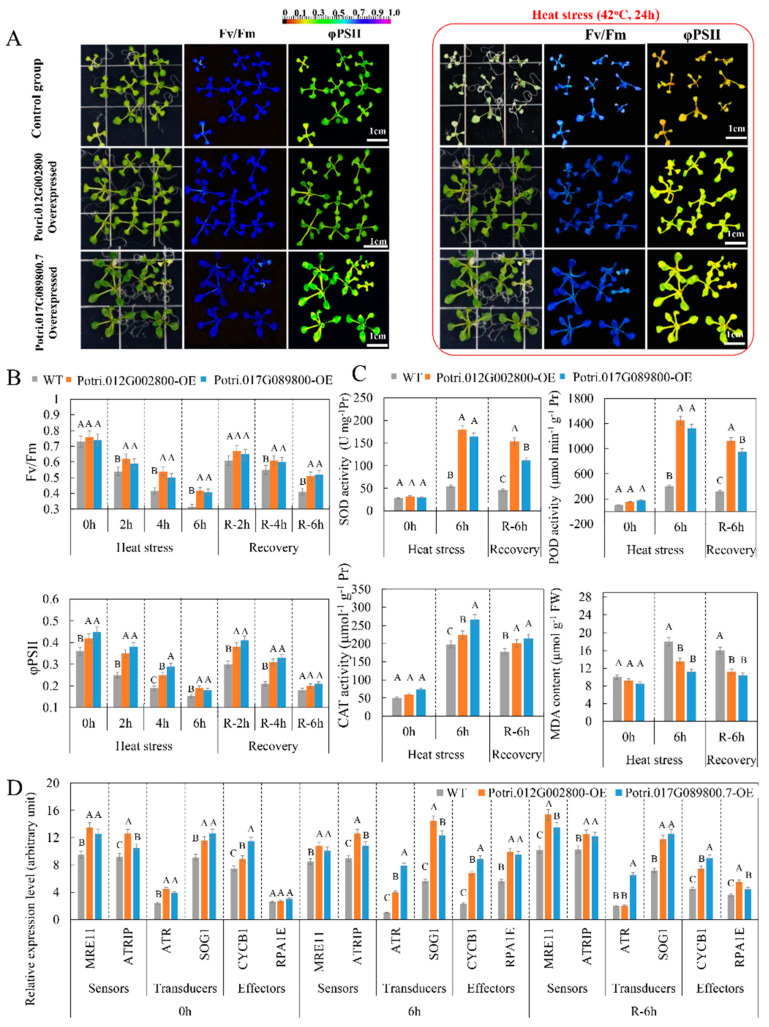
Chlorophyll fluorescence, physiological, and biochemical characteristics of transgenic *Arabidopsis*: (**A**) chlorophyll fluorescence imaging of WT and transgenic *Arabidopsis* under heat stress (14 days old, 42 °C, 24 h) and (**B**) chlorophyll fluorescence parameters of the WT and transgenic *Arabidopsis* under heat stress (14 days old, 42 °C, 2, 4 and 6 h) and recovery. Fv/Fm represents the Fv/Fm ratio, i.e., the maximum efficiency of photosystem II (PSII) photochemistry; ΦPSII represents the quantum yield of PSII electron transport. (**C**) Change of superoxide dismutase (SOD), peroxidase (POD), and catalase (CAT) activities and malondialdehyde (MDA) content in transgenic *Arabidopsis* under heat stress (42 °C, 6 h). (**D**) Expression pattern of DNA damage-related genes in transgenic *Arabidopsis* under heat stress: Activities are presented as mean ± standard error (*n* = 3). The data are mean ± SD for three separate treatments. Significant differences in the physiological and biochemical characteristics and expression level among WT, Potri.017G089800-OE, and Potri.012G002800-OE in response heat stress were determined using Fisher’s LSD test. Different letters on error bars indicate significant differences at *p* < 0.01.

**Table 1 ijms-21-06808-t001:** Candidate lncRNA and their targets genomic location and annotation.

Locus Name	Location	Targets	Annotation
TCONS_00260893	Chr17:10685952-10690606 (Antisense strand)	Potri.017G089800	Cyclic nucleotide- regulated ion channel family protein
TCONS_00202587	Chr12: 251176-251434 (Sense strand)	Potri.012G002700	Highly ABA-induced PP2C gene 3
		Potri.012G002800	Protein kinase superfamily protein

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
