# Peer review of "High-Temperature-Responsive Poplar lncRNAs Modulate Target Gene Expression via RNA Interference and Act as RNA Scaffolds to Enhance Heat Tolerance"

_ijms, 2020, doi:10.3390/ijms21186808_

Round 1
Reviewer 1 Report
The authors in this manuscript have presented a study showing the role of lncRNAs in heat-response / modulations in Poplar. Overall I am happy with the quality of the paper. I have a few observations:
I did notice that when abbreviation were introduced in the "Introduction" section of the manuscript, the expansions were not provided. For example, line 95 introduces PSII without mentioning what it stands for. Even though the material and methods explain these, it will be best to have it expanded when introduced first in the manuscript.
Line 106 introduces the RNA-Seq experiments. The number of reads are listed in line 109 however it doesnt mention how many replicates / read counts per replicate. It will be best if this is clarified.
Can the authors clarify the difference between Table S2 and S4 ?
Few minors typos need to be checked. For example line 298 there needs a space between "TCONS_00260983" and "and"
The authors have introduced terms like CNGCs and DDR without providing the expansion of the abbreviations. This needs to be corrected.
Author Response
Reviewer#1
- I did notice that when abbreviation were introduced in the "Introduction" section of the manuscript, the expansions were not provided. For example, line 95 introduces PSII without mentioning what it stands for. Even though the material and methods explain these, it will be best to have it expanded when introduced first in the manuscript.
Our response: We thank Reviewer for this constructive suggestion. The interpretation of the abbreviation has been added in current manuscript.
For instance:
“To identify high temperature-responsive lncRNAs in the leaves of P. simonii, the stability of the photochemical efficiency of photosystem II (PSII) and antioxidant enzyme activity were used to optimize the high-temperature treatment.”
- Line 106 introduces the RNA-Seq experiments. The number of reads are listed in line 109 however it doesnt mention how many replicates / read counts per replicate. It will be best if this is clarified.
Our response: We thank Reviewer for this constructive suggestion. The details of replicates/ read count per replicate has been added in Table S1.
For instance:
“Sequencing of cDNA libraries generated from control and heat-treated leaf samples yielded approximately 156 and 149 million high-quality reads (mean values of three replicates), respectively. Of these, about 79% and 81% of reads from the control and heat-treated samples, respectively, could be mapped to the reference genome (Table S1).”
|
Table S1 Statistics of mapping results |
||||
|
lncRNA |
  |
Clean reads |
Mapped reads |
Mapping rate |
|
  |
||||
|
Control group |
Replicates 1 |
154,890,760 |
113,884,137 |
78.60% |
|
Replicates 2 |
158,995,200 |
123,682,366 |
77.79% |
|
|
Replicates 3 |
156,622,796 |
132,487,223 |
84.59% |
|
|
Heat stress |
Replicates 1 |
145,448,012 |
120,285,505 |
82.70% |
|
Replicates 2 |
158,852,636 |
131,736,491 |
82.93% |
|
|
Replicates 3 |
144,349,738 |
114,685,866 |
79.45% |
|
(3) Can the authors clarify the difference between Table S2 and S4?
Our response: We thank Reviewer for this constructive suggestion. All of genes which detected in control or heat stress libraries were store in Table S2. Only different expressed genes (fold change > 2 or < 0.5, p < 0.01) between control group and heat stress were store in Table S4.
(4) Few minors typos need to be checked. For example line 298 there needs a space between "TCONS_00260983" and "and"
Our response: We thank Reviewer for this constructive suggestion. It has been revised in the current manuscript.
For instance:
“TCONS_00260893 has been predicted to target Potri.017G089800, which encodes cyclic nucleotide-gated ion channel 2 (CNGC2) that also known and important signal transduction gene. TCONS_00260893 and Potri.017G089800 exist as a partially overlapping natural antisense transcript (NAT).”
(5) The authors have introduced terms like CNGCs and DDR without providing the expansion of the abbreviations. This needs to be corrected.
Our response: We thank Reviewer for this constructive suggestion. The interpretation of the abbreviation has been added in current manuscript.
For instance:
“Over the past two decades, cyclic nucleotide-gated ion channels (CNGCs) have been extensively studied in plants.”
“Among the 185 genes, 3 and 2 were found to be involved in cell death and the cell cycle, respectively, indicating that heat-responsive lncRNAs might be involved in the DNA damage response (DDR) through regulation of the expression of these genes (Table S7).”

Reviewer 2 Report
Dear Editor,
The current manuscript presents some very interesting observations. I am very satisfied with the current form and I would recommend accept
Author Response
Thank you very much for your positive comments.
Reviewer 3 Report
The study revealed the role of the lncRNAs in regulating their target genes. I have a few comments to improve the manuscript for publication:
Line 22: “regulation” instead of “regulate”
Line 117: please filter all lncRNAs shorter than 100bp
Lines 117 – 120: how did you determine the length range of most of the genes?
Line 121: “the sense lncRNA loci”; do the authors mean sense protein-coding genes?
Line 136: replace “predicate” with “predict”
Line 498: “The lncRNAs were sequenced”; If total RNA was sequenced, why did not you mention mRNA as well or say total RNA?
Author Response
"Please see the attachment.
